# Bifurcations and bursting in the Epileptor

**Maria Luisa Saggio**✱, **Viktor Jirsa**

Institut de Neurosciences des Systemes INS UMR1106, AMU, INSERM, Marseille, France

\* maria-luisa.saggio@univ-amu.fr

## Abstract

The Epileptor is a phenomenological model for seizure activity that is used in a personalized large-scale brain modeling framework, the Virtual Epileptic Patient, with the aim of improving surgery outcomes for drug-resistant epileptic patients. Transitions between interictal and ictal states are modeled as bifurcations, enabling the definition of seizure classes in terms of onset/offset bifurcations. This establishes a taxonomy of seizures grounded in their essential underlying dynamics and the Epileptor replicates the activity of the most common class, as observed in patients with focal epilepsy, which is characterized by square-wave bursting properties. The Epileptor also encodes an additional mechanism to account for interictal spikes and spike and wave discharges. Here we use insights from a more generic model for square-wave bursting, based on the Unfolding Theory approach, to guide the bifurcation analysis of the Epileptor and gain a deeper understanding of the model and the role of its parameters. We show how the Epileptor's parameters can be modified to produce activities for other seizures classes of the taxonomy, as observed in patients, so that the large-scale brain models could be further personalized. Some of these classes have already been described in the literature in the Epileptor, others, predicted by the generic model, are new. Finally, we unveil how the interaction with the additional mechanism for spike and wave discharges alters the bifurcation structure of the main burster.

**Data Availability Statement:** All the relevant data are in the manuscript and it Supporting information files.

**Funding:** VJ received fundings through the EU's Horizon 2020 Framework Programme for Research and Innovation under the Specific Grant

## Author summary

This work focuses on a model, the Epileptor, which mimics seizure activity observed in many patients with focal epilepsy. The Epileptor forms the core of a large-scale brain modeling framework, known as the Virtual Epileptic Patient. This framework is developed to provide clinicians with a platform for testing outcomes of virtual surgeries and stimulations, and for finding optimal strategies to halt seizures in drug-resistant epileptic patients. However, models of brain activity, such as the Epileptor, can contain a large number of parameters, making it a daunting task to appreciate the full range of behaviors they can produce. Here, we approach this problem using a simpler model as a guide. The simpler model provides a map to orient ourselves, allowing us to identify key parameters for investigation and suggesting how to tweak them to obtain a broader range of behaviors. This is important because this repertoire of behaviors includes different classes of

Agreement number 945539 (Human Brain Project SGA3 https://www.humanbrainproject.eu/en/science-development/scientific-achievements/deliverables/third-specific-grant-agreement/) and through the French National Research Agency as part of the second Programmed Investissements d'Avenir (ANR-17-RHUS-0004, EPINOV https://clinicaltrials.gov/study/NCT03643016). VJ and MLS received fundings from the European Union's Horizon Europe Programme under the Specific Grant Agreement No. 101147319 (EBRAINS 2.0 Project). The funders had no role in study design, data collection and analysis, decision to publish, or preparation of the manuscript.

**Competing interests:** The authors have declared that no competing interests exist.

seizures that can be used to make the Virtual Epileptic Patient framework more patient-specific and potentially increase its predictive power.

## Introduction

Epilepsy is the most common among the chronic and severe neurological diseases, affecting 65 million people worldwide [1] and is characterized by an augmented susceptibility to seizures. The complexity of this group of diseases unfolds along different axes. The complexity comprises the multifactorial causes, since 'almost any condition affecting the cerebral grey matter can result in epilepsy' [2], and the presence of interacting processes spanning several scales in time and space. From mechanisms producing very fast 2kHz oscillations to ultra-slow processes spanning the entire lifespan of an individual, from within neuron processes to whole brain network mechanisms, with a variety of brain structures that can be involved depending on the type of epilepsy but also in a patient-specific manner. As nicely state already in 1874 by J. H. Jackson [3]:

> In any epilepsy, there is but 'one cause' physiologically speaking—viz. the instability of grey matter, but an unknown number of causes if we mean pathological processes leading to that instability.

This translates in the striking fact that, despite the described heterogeneity, seizures' electrographic signatures are remarkably stereotyped across patients [4], different animal species and primitive laboratory models [5]. This limited variability, together with epilepsies being dynamical diseases characterized by abnormal dynamics [6–9], has been proposed to reflect the existence of invariant dynamical properties in seizures' underlying mechanisms that could result from a variety of biological processes.

Models that focus on reproducing relevant features observed in data, that is phenomenological models, are particularly useful in the study of epileptic seizures, because they allow us to focus on these essential dynamics without committing to a specific choice in terms of biological implementation. Insights from this class of models can complement but also guide our understanding of the dynamical repertoire of more complex biophysically inspired models, establishing a link between abstract dynamical and specific physiological mechanisms [10].

The identification of a series of abstract dynamical mechanisms characterizing seizures allows to add a new dimension to current clinical classification that is operational and based on phenotype [11]. There are a few initial proposals in this direction. Some exploits more detailed spatiotemporal dynamics. For example, spatiotemporal heterogeneities in the distribution of excitability within a single brain region produce either low amplitude fast activity (LAF) or high amplitude slow activity (HAS) at seizure onset through different dynamical mechanisms that correlate with surgical outcomes [12]. Spatiotemporal dynamics at different scales also provides dynamical mechanisms underlying synchronous or asynchronous seizure termination across brain regions [13]. Another, complementary, proposal neglects the spatial component within a brain region and assumes that the mesoscopic abrupt transitions between interictal and ictal states are brought about by bifurcations. Classes are defined as the onset/offset bifurcations pair that delimitates the seizure [5], extending to epilepsy a taxonomy that was previously developed by [14] for use in neuronal bursting. In the context of epilepsy, we call

these bursting classes 'dynamotypes' [15], establishing the taxonomy we refer to in the present work.

Identifying key dynamical characteristics that differentiate seizures could have several potential practical implications. Phenomenological models incorporating these key features could be used, for instance, to devise specific strategies to exit from the ictal state and abort a seizure, because different characteristics of a dynamical system require different types of stimulation to destabilize the oscillatory regime [16–20].

In addition, due to their low computational burden, phenomenological models for seizure activity are prime candidates to be used in whole-brain simulations. Such patient-specific models incorporate patient data, such as structural or functional data, to build large scale networks in which brain regions are endowed with a dynamical model and coupled together. Most studies using this approach in the context of epilepsy have focused on potential strategies to improve surgical outcome in drug-resistant patients. For some of these patients, an alternative to medication is the surgical resection of the brain regions involved in the generation of seizures, the epileptogenic zone (EZ), under the constraint of limiting post-surgical neurological impairments [21]. However, outcomes of this type of surgery are very variable and depend on the patient's condition and epilepsy, with surgery success rates ranging between 34% and 74% [22]. The inference of the EZ is highly non trivial and large-scale patient specific approaches aim at arming clinicians with an additional tool to use in their evaluation. This tool leverages on the possibility, through whole brain simulations, to reveal otherwise hidden complex network and dynamical effects. It can be used to test specific clinical hypotheses (e.g. 'will seizures stop by removing a specific set of brain regions?') by simulating functional data or to find unbiased optimal surgery strategies through the applications of different metrics or parameters fitting techniques. These models can contribute to different stages of the presurgical evaluation by studying seizure propagation, resection strategies, predicted surgery outcome or strategies to limit cognitive impairment [13, 23–37], or to improve presurgical implantation of intracranial electrodes [38]. Other proposed applications, beyond surgery, are related to the diagnosis of focal and generalized epilepsy [39], the evaluation of the increased proneness to seizures in Alzheimer's disease [40] and the investigation of the effects of brain stimulation [41].

The initial results from these retrospective studies have been promising, and one of these frameworks, the Virtual Epileptic Patient (VEP) [17, 27, 37], is currently being validated in a prospective clinical trial involving more than 300 patients in France. Patient-specificity is key and, under the requirement that state of the art methods are used in diffusion imaging, it has been shown that the patient-specific connectome gives the best outcome for the VEP [27]. At the brain region level, the Epilepor model is used for all patients. This model has been proposed by [5] to phenomenologically reproduce the most predominant dynamotype as observed in *in vitro* preparations, zebrafish, mice and human epileptic patients with focal seizures. These patients, though, exhibited other dynamotypes as well [5, 15]. The impact of the dynamotype on the VEP outcome has not yet been investigated, but we know that different classes may behave differently in terms of synchronization and propagation properties [16, 42]. Exploring the full potential of the Epileptor model in terms of bursting dynamics, and how to set the parameters in order to obtain different classes, is an important step to (i) improve our understanding of the model (ii) explore the possibility that a further personalization of the VEP model in term of patient-specific dynamotype could alter the model outcome.

The most common dynamotype identified in data from patients with focal seizures had Saddle-Node (SN) bifurcation for the onset of the fast oscillations and a Saddle-Homoclinic (SH) bifurcation for their offset [5]. The Epileptor phenomenologically incorporates this fast-

slow bursting dynamics, together with other forms of epileptiform activity, such as preictal spikes and spike-and-wave complexes.

It has been noted that a minimal model for this type of bursting (fast-slow SN/SH bursting, also known as square-wave bursting) can be obtained by using a layer of the unfolding of the degenerate Takens-Bogdanov (DTB) singularity to create a 'map' of possible behaviors and by adding a slow dynamics to promote movement on this map [43] and that this layer could host other types of bursters [44]. Building on this, we have developed a minimal model with a rich repertoire of classes from the taxonomy [45]. We will refer to it as the DTB bursting model.

In the present work we use insights from fast-slow bursters and the DTB bursting model to investigate the Epileptor model. In particular, we identify the fast parameters that contain the parametrization, in terms of slow variables, of the 'path' through which the fast subsystem is pushed. We explore the bifurcation diagram of these fast parameters ('the map') to identify all the relevant bifurcation curves we expect to find based on the knowledge of the unfolding of the DTB singularity [46]. We show how the Epileptor moves on the map during a seizure and how alternative paths with different onsets and offsets can be placed in this single map, some of them not yet observed in the Epileptor. We highlight how the input from the additional system for preictal spikes and spike-and-wave complexes alters the path on the map, and thus the sequence of bifurcation curves encountered by the system. Finally, we show the effect that the main Epileptor's parameter, the epileptogenicity $x_0$, that is the brain region proneness to seizures, has on the path.

## Models

### The DTB bursting model

The DTB bursting model [45] uses the unfolding of the DTB singularity as fast subsystem, this gives a map of the possible behaviors when the unfolding parameters ($\mu_1$, $\mu_2$, $v$) are modified. The portion of this map that is relevant for SN/SH bursting can be obtained by taking a layer of the unfolding for fixed positive $\mu_2$ or for fixed positive $v$. If the fixed parameter is chosen small enough, the maps obtained will be topologically equivalent [46]. Here we consider a layer for $\mu_2 = 0.07$, as shown in Fig 1A. There are regions with only one stable attractor, either a fixed point or a limit cycle (white) and regions of bistability between two fixed points (grey) or a fixed point and a limit cycle (yellow). These regions are separated by bifurcation curves: SN, SH, supercritical Hopf (SupH) and Saddle-Node-on-Invariant-Circle (SNIC).

If the two parameters ($\mu_1$, $v$) slowly change as a function of a variable $z$ we can have movement on the map. We parametrize ($\mu_1(z)$, $v(z)$) so that the allowed movements are straight lines:

$$\begin{cases} \mu_1 = \mu_{1,0} + d_{\mu1}z; \\ v = v_0 + d_v z; \end{cases} \tag{1}$$

with ($\mu_{1,0}$, $v_0$) being the initial point of the path and ($d_{\mu1}$, $d_v$) the direction vector of the path. We then impose a simple dynamics for $z$ such that: when the system is at or close to rest $z$ increases and the fast subsystem moves rightward on the map; when the system is far from rest $z$ decreases and the system moves leftward.

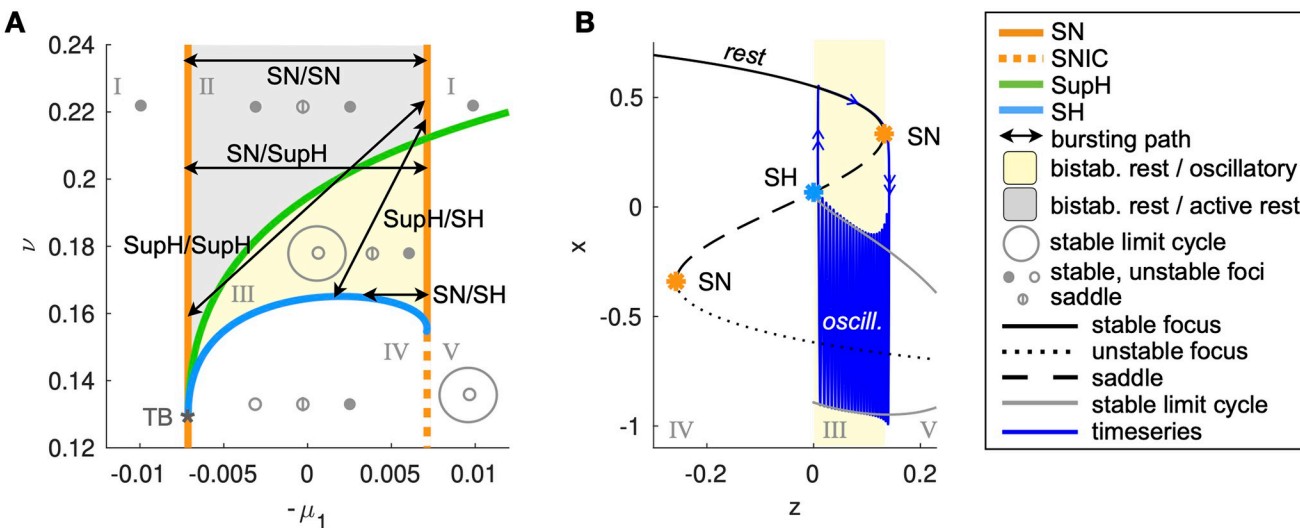

**Fig 1. Hysteresis-loop bursting in the DTB bursting model.** A: This is one portion of the unfolding in which SN/SH bursting can be placed, together with other classes [45]. Saddle-Node (SN) and supercritical Hopf (SupH) curves meet at the Takens-Bogdanov point TB. Bifurcation curves partition the map in five regions with different state space configurations (Roman Numerals). When this map is used for the fast subsystem of a fast-slow bursters with a hysteresis-loop mechanism for the slow variable, possible classes in the map are: SN/SH, SN/SupH, SupH/SH and SupH/SupH plus SN/SN where the system alternates between the two stable fixed points [45]. When more than one fixed point exist, the resting (or inter ictal) state is the one on the right, the other one we call 'active rest'. The resting state corresponds to the upper branch of fixed points in panel B. B: Typical bifurcation diagram for the SN/SH class. When the system is at rest, $z$ increases until the fixed point destabilizes through the SN bifurcation and the system jumps into the stable limit cycle. Now that the system is far from rest, $z$ decreases until the limit cycle destabilizes through a SH bifurcation and the system jumps back to rest. If the destabilization of the fixed point/limit cycle is obtained through a different bifurcation, we will have a different onset/offset class and a different appearance of the burster's timeseries.

The model's equations read:

$$\begin{cases} \dot{x} = -y \\ \dot{y} = x^3 - \mu_2 x - \mu_1(z) - y(v(z) + x + x^2) \\ \dot{z} = c(x - x_0) \end{cases} \quad (2)$$

where $c \ll 1$ gives the time separation among the two subsystem and $x_0$ sets the threshold for inverting the behavior of $z$. In this types of models $x_0$ is called 'excitability', because it expresses how prone the system is to move towards the destabilization of the resting state. In Eq (2) the dynamics of the slow variable has been simplified as compared to the original model, in a way that is only possible in some regions of the unfolding. See the Methods Section for more details.

If we initialize our fast subsystem inside the bistability region, when at rest it moves towards a SN curve that destabilizes the fixed point and forces the system to jump to the limit cycle (or to the other fixed point if in the grey region). Now the fast subsystem starts moving leftwards until it reaches a bifurcation that destabilizes the new attractor so that the fast subsystem jumps back to rest. This loop is what constitutes fast-slow hysteresis-loop bursting. Depending on where the path is placed on the map, the system will encounter specific sequences of bifurcation curves, so that different types of bursting are possible (Fig 1A).

Since this model is generic for SN/SH bursting, we can expect that a topologically equivalent map exists in the Epileptor model for some values of the parameters and that it is possible to adjust these parameters to have other bursting classes.

## The Epileptor

For the Epileptor model, we use equations as in [5], with the addition of the parameter $m$ as in [47]:

$$\dot{x}_1 = \begin{cases} y_1 - x_1^3 + 3x_1^2 - z + I_{rest_1} & \text{if } x_1 < 0 \\ y_1 + (m - x_2 + 0.6(z-4)^2)x_1 - z + I_{rest_1} & \text{if } x_1 \geq 0 \end{cases}$$

$$\dot{y}_1 = y_0 - 5x_1^2 - y_1$$

$$\dot{z} = \frac{1}{\tau_0}(4(x_1 - x_0) - z) \tag{3}$$

$$\dot{x}_2 = -y_2 + x_2 - x_2^3 + I_{rest2} + u - 0.3(z - 3.5)$$

$$\dot{y}_2 = \begin{cases} -\frac{1}{\tau_2}y_2 & \text{if } x_2 < -0.25 \\ \frac{1}{\tau_2}(-y_2 + 6(x_2 + 0.25)) & \text{if } x_2 \geq -0.25 \end{cases}$$

where

$$u = 0.002 \int_{t_0}^{t} e^{-\gamma(t-\tau)} x_1(\tau), d\tau \tag{4}$$

with $x_0 = -1.6$, $y_0 = 1$, $\tau_0 = 2857$, $\tau_2 = 10$, $\tau_1 = 1$, $I_{rest_1} = 3.1$, $I_{rest2} = 0.45$ and $\gamma = 0.01$.

The Epileptor is composed of three subsystems acting on different time scales: a fast, an intermediate and a slow subsystem.

A fast subsystem $(x_1, y_1)$, acting on the scale described by the time constant $\tau_1$, is based on a modified Hindmarsh and Rose model and is responsible for the fast oscillatory activity observed during a seizure. This system can display bistability between a stable fixed point (resting state interpreted as interictal condition) and a stable limit cycle (fast oscillatory activity interpreted as the ictal state), similarly to the yellow region in Fig 1A. As in the DTB bursting model, the slow variable $z$ allows for hysteresis-loop bursting in this region (Fig 2, blue). This mechanism is the core of the Epileptor model.

In addition, subsystem $(x_2, y_2)$ acts on an intermediate time scale, given by $\tau_2$, and is an excitable system with a SNIC bifurcation. It is responsible for the generation of preictal spikes, which reflect the increased excitability of this subsystem close to seizure onset, and for spike and wave complexes during the seizure. It displays 'slow-wave' SNIC/SNIC bursting (Fig 2, magenta) in which the role of slow variable is played by $u$, a low-pass filtered input from $x_1$ (see Eq (4)), combined with $z$. In this type of bursting there is no need for bistability nor feedback between the faster (i.e. the intermediate system) and slower variables [14]: independently of what the intermediate variable is doing, the slow variable pushes it back and forth across a SNIC bifurcation. The activity of the intermediate subsystem modulates that of the fast one.

The Epileptor field potential (Fig 2, purple) is given by a combination of fast and intermediate variables: $x_2 - x_1$.

## Results

Given the presence of time scale separation, with $\tau_1 \ll \tau_2 \ll \tau_0$, the three subsystems can be analyzed in isolation [48].

We here focus on the core hysteresis-loop mechanism. To do so we will study the relevant parameter space of the fast subsystem (this subsystem is the one containing the bifurcations

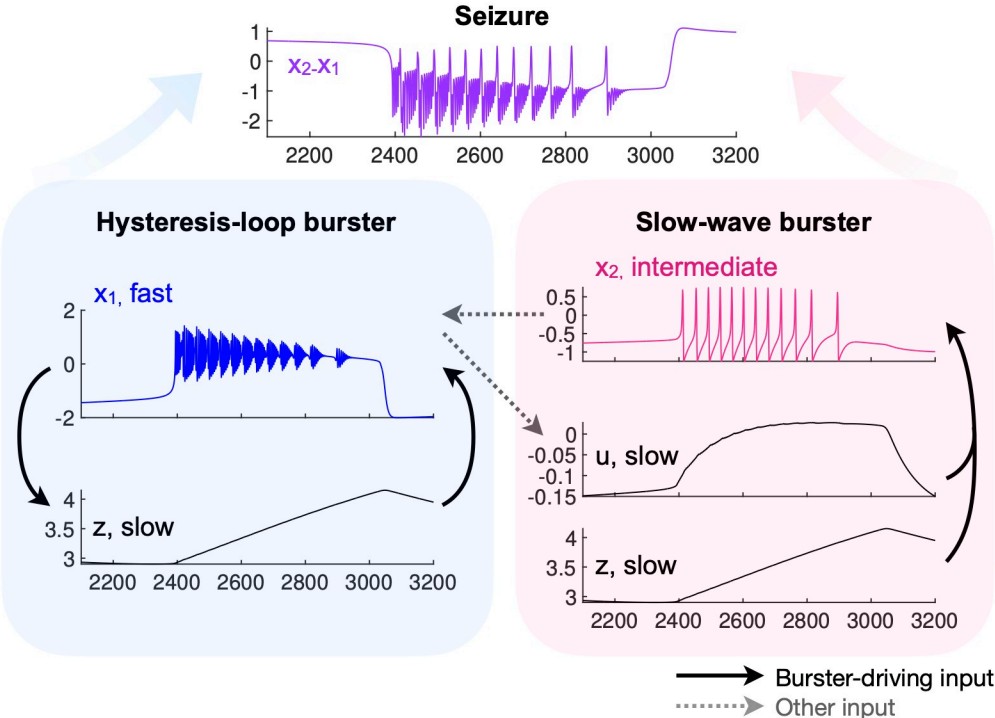

**Fig 2. Epileptor's main mechanisms.** The Epileptor field potential (purple), is a combination of the activity of one fast variable (blue) and an intermediate one (magenta). The fast and slow variables dynamics and feedback among them constitute the core of the model (blue), while the intermediate variables (magenta) modulate the fast activity. All the simulations in this work are performed without noise.

for the onset and offset of seizures) to create a 'map' of its dynamical repertoire, guided by what we know about the map of the DTB bursting model. We will then follow the path of the full Epileptor model in this map.

## Fast subsystem—The map

In this section we define, for better readability $(x_1, y_1) = (x, y)$. We are interested in whether different types of behaviors can be produced when the fast parameters are allowed to vary. We thus rewrite the fast subsystem in Eq (3), to highlight the presence of the parameters of the model, as:

$$\dot{x} = \begin{cases} vy + ax^3 + bx^2 + \mu & \text{if } x < 0 \\ vy + \bar{m}x + \mu & \text{if } x \geq 0 \end{cases}$$

$$\dot{y} = y_0 + Bx^2 + Ny \tag{5}$$

Here we consider $v, a, b, \mu, \bar{m}, y_0, B$ and $N$ as parameters of the fast subsystem, and interpret

$$\mu(z) = I_{rest} - z$$

$$\bar{m}(z, x_2) = 0.6\alpha(z-4)^2 + m - x_2 \tag{6}$$

as a parametrization of the path that the fast subsystem follows in its parameter space as promoted by feedback from the slower variables $z$ and $x_2$.

The fast subsystem as in Eq (3) can be obtained from Eqs (5) and (6) by replacing: $v = 1$, $a = -1$, $b = 3$, $\mu = I_{rest} - z$, $B = -5$, $N = -1$ and $\alpha = 1$.

Since $(\mu, \bar{m})$ are the fast parameters that slowly change in the Epileptor's dynamics, we expect that, when they are treated as bifurcation parameters, they will produce a map similar to that of the generic model. This entails, beside curves for the onset (SN) and offset (SH) bifurcations, the presence of a curve of H bifurcation that encounters the SH curve on a SN curve different than the onset one.

In all the figures and simulations we set parameters other than $(\mu, \bar{m})$ as in Eq (3), unless otherwise specified, except for $x_0 = -2$, which we have modified for visualization purposes. As explained later, this doesn't alter the onset/offset pattern. However, in the bifurcation analysis, we will maintain all the parameters in order to gain some insights on their role in shaping the map.

**Bifurcation analysis. Fixed points**. The fixed points of the system can be obtained by imposing:

$$
\begin{cases}
\dot{y} = 0 \iff y = -\frac{y_0 + Bx^2}{N} \\
\dot{x} = 0 \iff \begin{cases} ax^3 + \left(b - \frac{vB}{N}\right)x^2 + \left(\mu - \frac{vy_0}{N}\right) = 0 & \text{if } x < 0 \\ -\frac{vB}{N}x^2 + \bar{m}x + \left(\mu - \frac{vy_0}{N}\right) = 0 & \text{if } x \geq 0 \end{cases}
\end{cases}
\tag{7}
$$

The solutions are such that, in the $(\mu, \bar{m})$ space, there is a central region with three fixed points, while outside this region there exist only one solution: the lower branch of the fixed points manifold for small values of $\mu$ and the upper branch for big values (Fig 3).

**Saddle-Node manifolds**. The central region with three solutions is delimited by two curves of SN bifurcations, satisfying the additional condition that the determinant of the Jacobian of the system is zero:

$$
J = \begin{pmatrix} \frac{\partial \dot{x}}{\partial x} & \frac{\partial \dot{x}}{\partial y} \\ \frac{\partial \dot{y}}{\partial x} & \frac{\partial \dot{y}}{\partial y} \end{pmatrix} = \begin{pmatrix} \begin{cases} 3ax^2 + 2bx & \text{if } x < 0 \\ \bar{m} & \text{if } x \geq 0 \end{cases} & v \\ 2Bx & N \end{pmatrix}
\tag{8}
$$

$$
\Delta = 0 \iff \begin{cases} x\left(3aNx + 2N\left(b - \frac{vB}{N}\right)\right) = 0 & \text{if } x < 0 \\ x = \frac{\bar{m}N}{2vB} & \text{if } x \geq 0 \end{cases} \iff \begin{cases} x = -\frac{2\left(b - \frac{vB}{N}\right)}{3a} & \text{if } x < 0 \\ x = \frac{\bar{m}N}{2vB} & \text{if } x \geq 0 \end{cases}
\tag{9}
$$

This gives a curve for negative values of $x$, $SN^-$ and a curve for positive values of $x$, $SN^+$. When $\bar{m} = 0$ the positive portion of the curve ends because $x = 0$. It joins, however, with a curve we find for $x = 0$, where, for $\bar{m} \leq 0$ the positive and negative branches of fixed points merge. This is not a real SN bifurcation curve because it occurs where the system is piece-wise and the positive and negative limits of the Jacobian do not match. However, for the goal of bursting it will behave as a SN curve. With these caveats we call it $SN^0$.

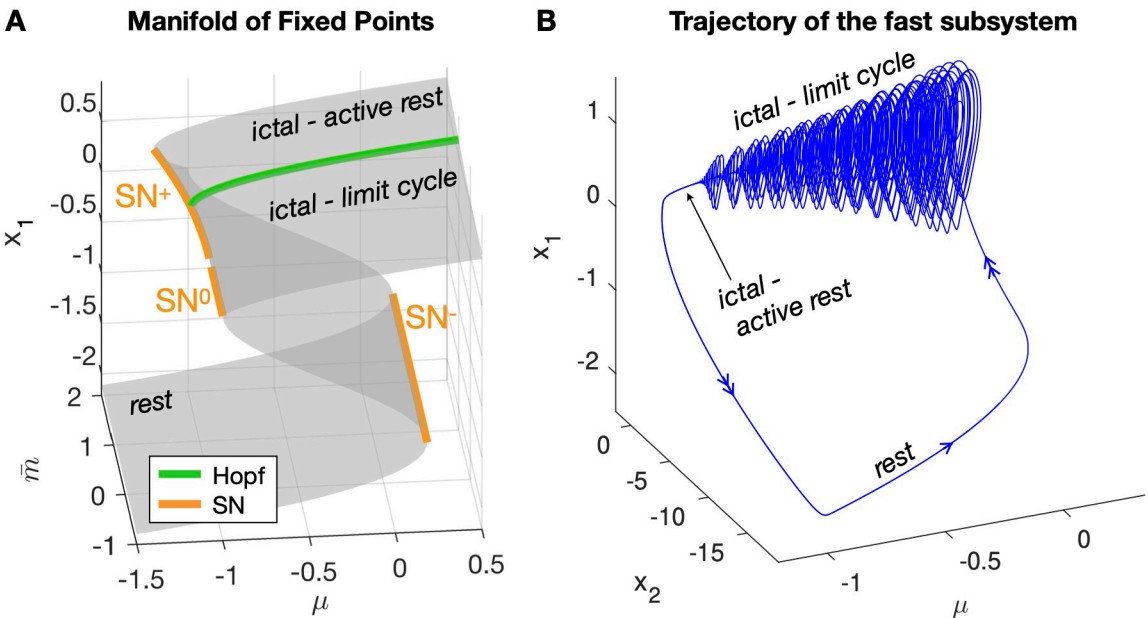

**Fig 3. Epileptor's fixed points and bursting trajectory.** A: Fixed points manifold for $x_1$. For $x_1 < 0$, there are two solutions, while for $x_1 \geq 0$ we have one fixed point when $\mu > \frac{vy_0}{N} = -1$ and two solutions for smaller values. The lower branch of fixed points is the rest or interictal state. The upper branch is stable for values of $\bar{m} > -N = 1$ and unstable otherwise. $SN^+$ occurs for $\mu = -1 - \frac{\bar{m}^2}{20}$, $SN^-$ for $\mu = \frac{5}{27}$, $SN^0$ for $\mu = -1$ and the Hopf curve for $\bar{m} = 1$. B: Trajectory of the Epileptor's fast subsystem $(x_1, x_2)$ plotted against the parameter $\mu$. It can be observed the hysteresis-loop caused by the two $SN$ curves.

By inserting the solutions in Eq (9) in Eq (7) we can find conditions on the parameters for the $SN^-$ and $SN^+$ curves. Imposing $x = 0$ in Eq (7) we find that one for $SN^0$:

$$
\begin{aligned}
SN^- : \mu &= \frac{vy_0}{N} - \frac{4}{27a^2}\left(b - \frac{vB}{N}\right)^3 \\[2mm]
SN^+ : \mu &= \frac{vy_0}{N} - \frac{N\bar{m}^2}{4Bv} \\[2mm]
SN^0 : \mu &= \frac{vy_0}{N}
\end{aligned}
\tag{10}
$$

**Andronov-Hopf manifold**. Beyond being fixed points, candidate Hopf bifurcation points must satisfy the condition that the trace of the Jacobian should be equal to zero:

$$
\tau = 0 \iff
\begin{cases}
x = \frac{-2b \pm \sqrt{4b^2 - 12aN}}{6a} & \text{if } x < 0 \\[2mm]
\bar{m} = -N & \text{if } x \geq 0
\end{cases}
\tag{11}
$$

When inserting Eq (11) in Eq (7) we find no solutions for $x < 0$. For $x \geq 0$, the condition $\bar{m} = -N$ gives $\Delta > 0$, and thus a Hopf curve [49], only on the upper branch of fixed points as shown in Fig 3A.

**Codimension 2 Takens-Bogdanov manifold**. We find a manifold of codimension 2 TB bifurcations where the Hopf and $SN^+$ merge, that is for $x = \frac{\bar{m}N}{2vB}$.

**Saddle-Homoclinic curve**. Once identified the local bifurcations, we searched for the SH curve. Since this is a global bifurcation, we used numerical tools and we performed our

analysis fixing all parameters as in [5] except for those used for the map. It was problematic to locate it using continuation softwares (Matcont) possibly due to the system being piece-wise. We thus performed simulations for different values of the parameters $(\mu, \bar{m})$, initializing the system close to the upper fixed point (either active rest or unstable fixed point) and computing the amplitude and frequency of the limit cycle with regards to the *x* variable (Fig 4 right panels). The trend of the frequency behavior is compatible with the presence of a SH bifurcation, which requires the frequency to scale to zero towards the bifurcation point. This curve stems from the TB point, as predicted from the theory, and reaches the other SN curve as in the unfolding of the DTB singularity. We have thus found a region of topological equivalence between the two models (Fig 4).

**Role of the other parameters on the topology of the map**. We can appreciate the role of some parameters, other than $(\mu, \bar{m})$, in shaping the map. For example, in Eq (10) we can see how they contribute to the SN curves. In particular, $y_0$ pushes all the curves right or left on the map, while *b* and *a* only act on the negative branch. With regards to the position of the Hopf curve, it simply depends on *N*. We can't make similar considerations for the SH curve, that we obtained numerically, except that the location of its starting point will depend on the location of the TB point where the H and the $SN^+$ curves meet.

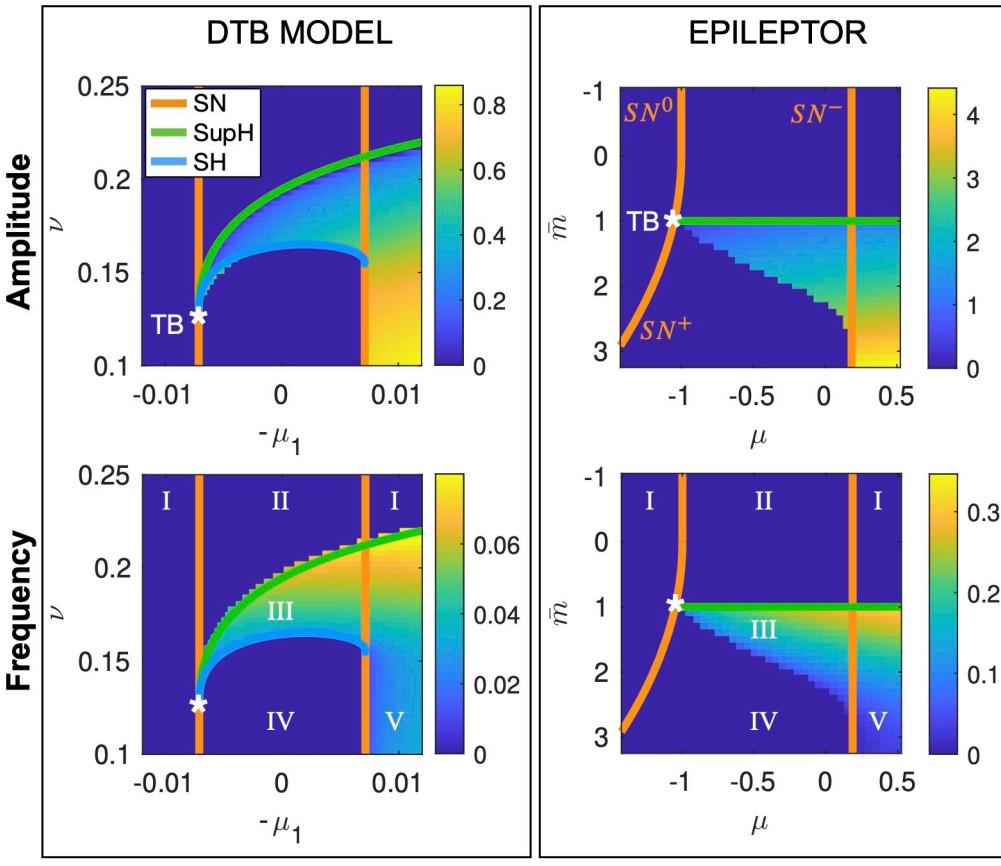

**Fig 4. Topological equivalence between bifurcation diagrams of the fast subsystems of the Epileptor and of the DBT bursting model.** In the left panels we show the portion of bifurcation diagram of the DTB bursting model (that is the unfolding of the DTB singularity) in which SN/SH bursting occurs, and the behavior of amplitude and frequency of the limit cycle. In the right panels, the same for the Epileptor's fast subsystem. In the latter case the presence of a SH bifurcation stemming from the TB point can be inferred by the behavior of the frequency (Hz) of the limit cycle identified through simulations, which scales down to zero. Roman Numerals refer to the configurations as in Fig 1.

Of course, these are considerations that only hold for small changes of the mentioned parameters. Understanding the intervals in which these parameters could be changed without altering the topology of the map would give information about its robustness. While useful, given that the physiological correlates of these parameters could fluctuate, this is beyond the scope of this work.

## Slower variables—Paths on the map

Now that we have the bifurcation diagram for the slowly changing parameters of the fast subsystem, we can go back to the path followed by the fast subsystem on this map under the influence of the fast and intermediate variables. The path is parametrized as in Eq (6), which we rewrite here for convenience:

$$
\begin{aligned}
\mu(z) &= I_{rest} - z \\
\bar{m}(z, x_2) &= 0.6\alpha(z - 4)^2 + m - x_2
\end{aligned}
\tag{12}
$$

**Placing on the map classes from the literature.**   Movement parallel to the $\mu$ axes is promoted by $z$ and is the core of the hysteresis-loop bursting, in which the slow variable, with feedback from $x_1$, pushes the fast subsystem across the onset and offset bifurcations. By changing the parameter $m$ we can move the path upward or downward, so that it will cross different pairs of bifurcation curves. In Fig 5, we plotted on the map the simulated paths for values of $m$ from the literature [5, 47] and show how they produce bursting of different classes, namely SN/SupH, SN/SH and SN/SN (in the latter burster, the onset/offset do not refer to oscillations, but simply to the alternation between the fixed points in the upper or lower branches). Of note, for $m = 0$ as in the original [5], input from $x_2$ is such that the actual offset of the hysteresis-loop burster is a SupH rather than a SH (Fig 5A). The SH offset can be retrieved by setting $I_{rest2} = 0$ so that there is no bursting in the intermediate subsystem (not shown).

For visualization purposes, we zoomed in on the seizure in the timeseries. However, these are periodic solutions, with the slow variable pushing the system along the closed paths on the map.

**Intermediate subsystem modulatory effects.**   While the fast subsystem is in the ictal state (either oscillating or in the active rest), its low-pass filtered activity and input from $z$ cause oscillations in the intermediate subsystem through a SNIC bifurcation. The intermediate fluctuations of $x_2$, in turn, modulate the $\bar{m}$ component of the path, and appear in Fig 5A–5C as 'spikes'. These spikes can cause the fast subsystem to cross the Hopf curve multiple times while moving towards seizure's offset. This is visible in the amplitude changes in the timeseries, as the amplitude decrease when approaching this bifurcation (Fig 5A and 5B). For example, let's consider the timeseries in Fig 5A. In the first part of the ictal state the fast subsystem briefly crosses the H curve but not long enough to settle down to the fixed point, while in the latter part of the timeseries the sequence of SupH bifurcations becomes more evident.

These 'spikes' in the path come from the activity of the intermediate subsystem, implying they are present even when $m$ is chosen such as to bring the path fully above the bistability region as in Fig 5C. However, since in this region no fast oscillations are possible, their modulatory effect is negligible. Finally, by setting $I_{rest2} = 0$ so that the intermediate subsystem doesn't reach the threshold for SNIC, as in Fig 5D, the spikes are no longer present in the path. We show it for $m = -8$. In this class, the active state of the Epileptor has been suggested to be linked to depolarization block [47], a physiological state in which action potentials cannot be triggered despite the neuronal membrane being depolarized.

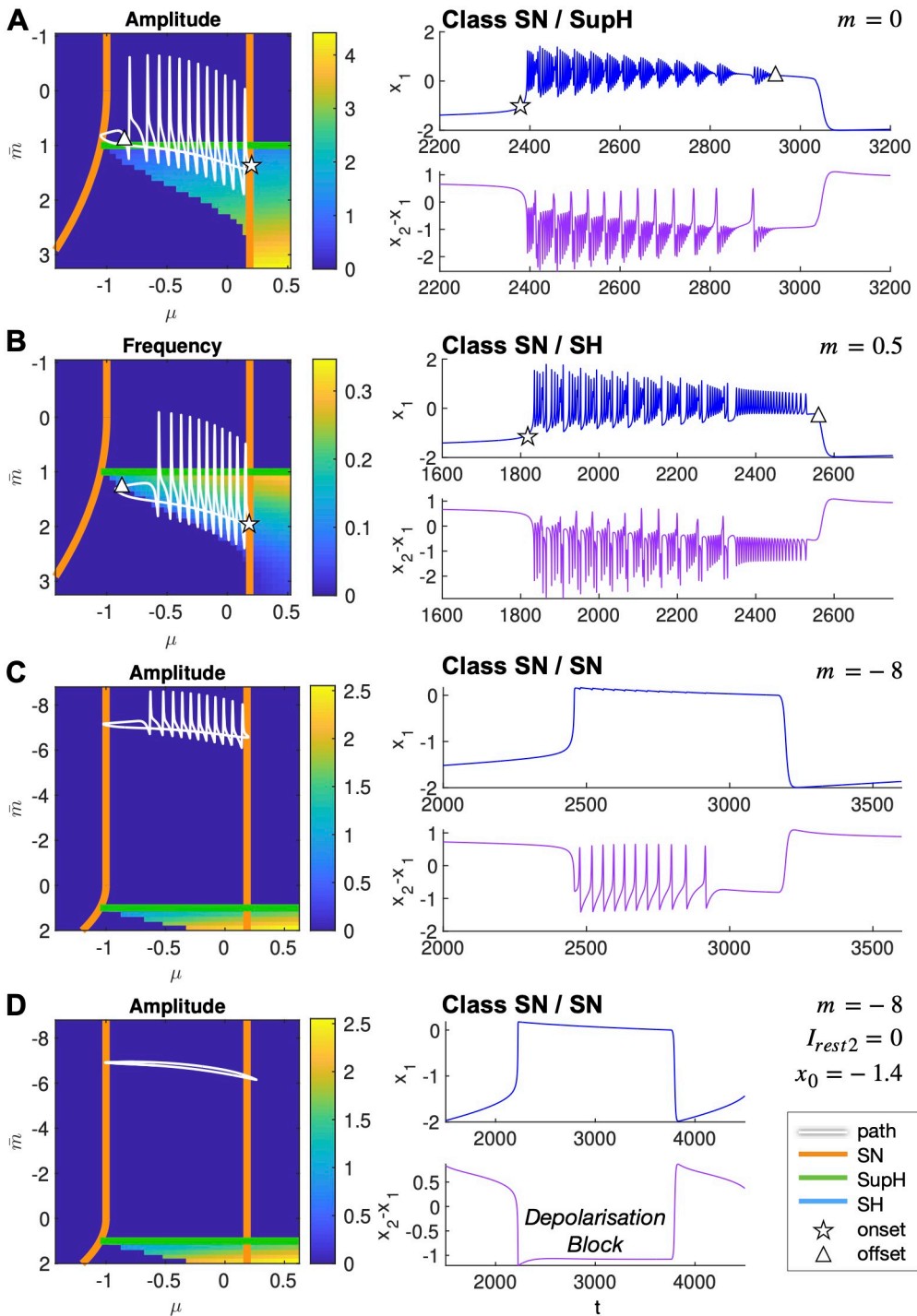

**Fig 5. Paths on the map.** Paths followed by the full Epileptor model in the map for four different conditions (left column), and the related timeseries (right column). All parameters are kept the same except for those specified. Both on maps and in timeseries, a star/triangle approximately marks the onset/offset of oscillations in the fast subsystem, when present. These types of Epileptor behaviors have been described in the literature.

**Identifying new classes of hysteresis-loop bursting.** The DTB bursting model allows for other two classes of hysteresis-loop bursting to be found in this map: SupH/SH and SupH/SupH (Fig 1A). By looking at Fig 5, we can observe that, in the Epileptor model, they can't be obtained by a simple vertical translation of the path. However, by changing the global slope of the path setting $\alpha = -1$ in Eq (6) and changing $m$ to appropriate values, we could simulate these other two classes as shown in Fig 6.

**Role of $x_0$ on the path.** With regards to the slow variable dynamics (Eq (3)), the relevant parameter is $x_0$, that is linked to how close the z-nullcline is to the resting state (Fig 7A). As in the DTB bursting model, the closer the z-nullcline is to the resting state, the more slowly $z$ evolves when at rest towards the ictal state and the faster it evolves when in the ictal state towards seizure offset (Fig 7B left, middle). When the z-nullcline crosses the branch of resting state, the intersection is a fixed point for the whole system (Fig 7B right) [47]. In the VEP model, $x_0$ is used to tune each brain region's epileptogenicity, that is its proneness to start a seizure. When considering only the hysteresis-loop mechanism, this parameter doesn't affect the path on the map, but rather the speed of movement along different parts of it. However, when the intermediate subsystem is in the oscillatory regime, different values of $x_0$ allow for a different amount of intermediate spikes along the path (Fig 7C).

Even though networks of coupled Epileptors are beyond the scope of this work, we want to point out that Epileptors are usually coupled through a fast-to-slow 'permittivity' coupling [50]. A region's proneness to seizures can be altered by incoming inputs from other brain regions, while the fast parameters, and thus the class, are unaffected.

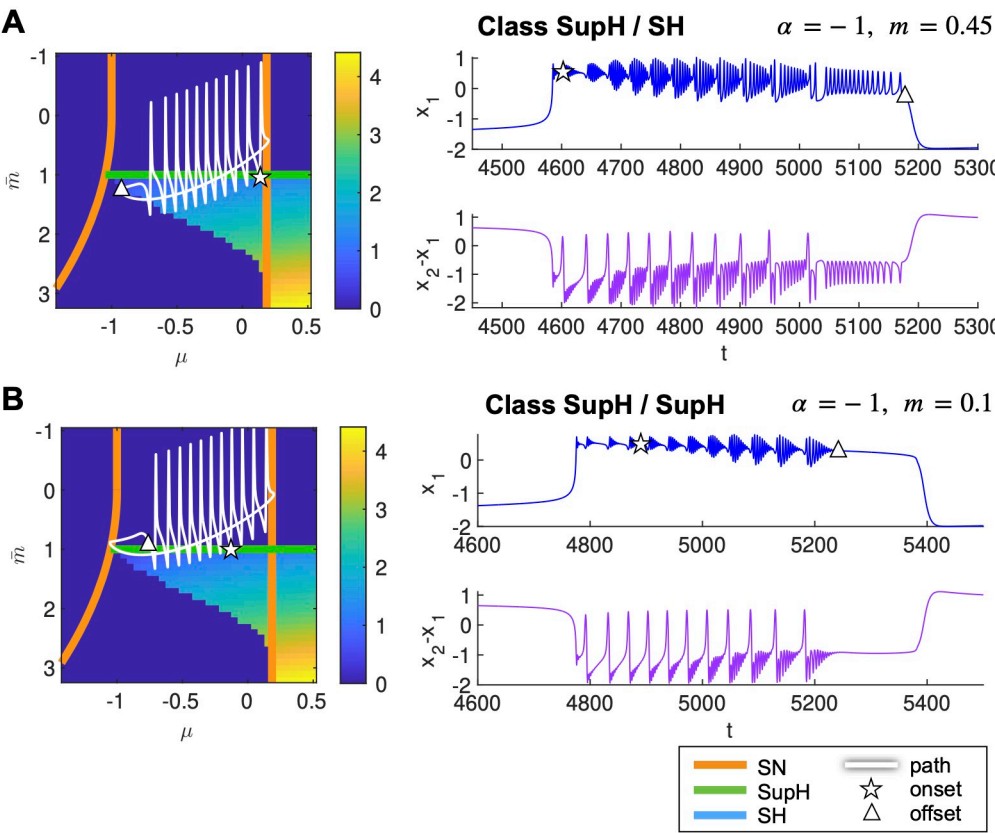

**Fig 6. New classes of bursting identified in the Epileptor model.** Path on the map (left) and timeseries (right) for the new classes SupH/SH (A) and SupH/SupH (B). Maps represent the amplitude of the limit cycle.

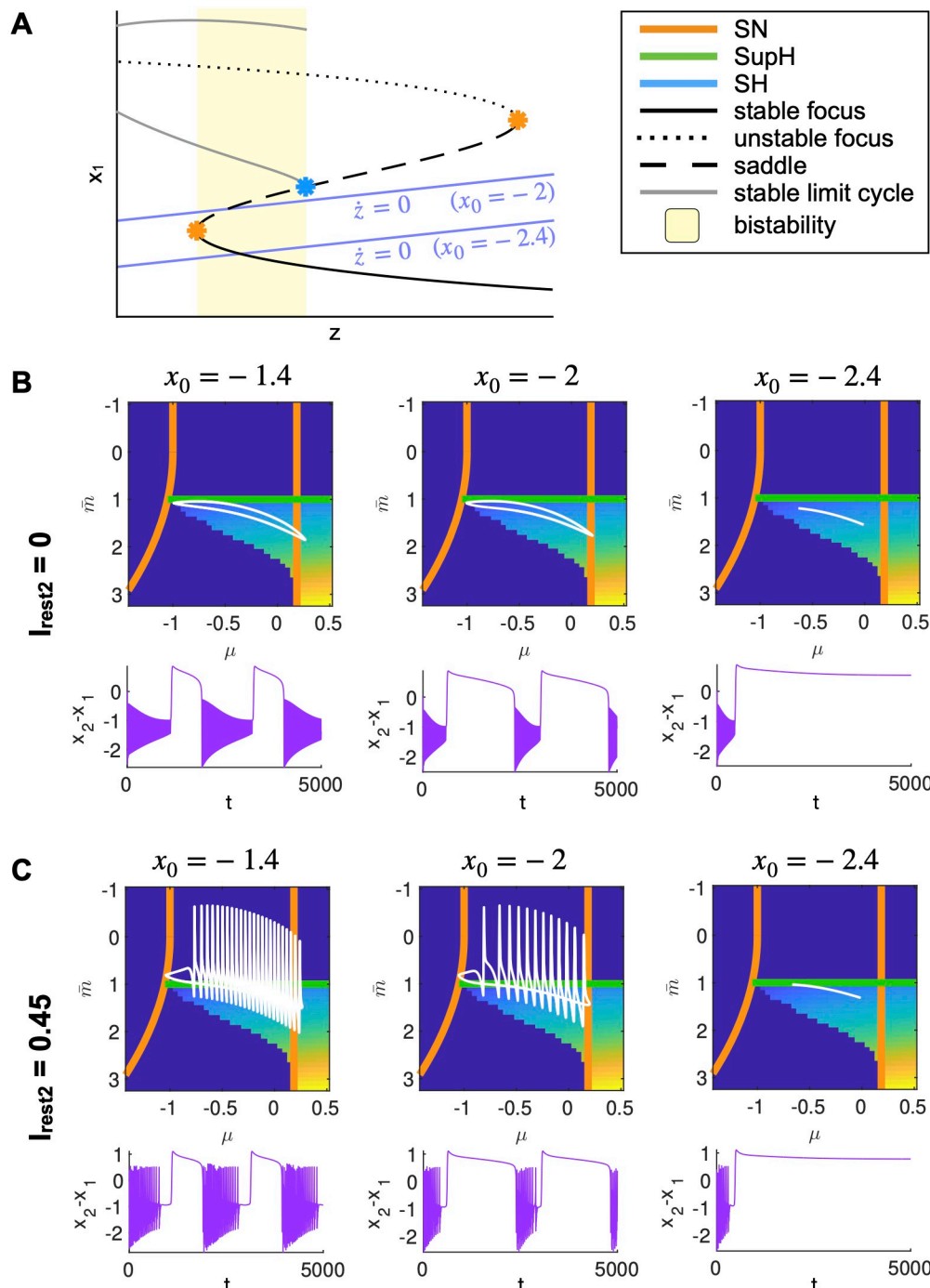

**Fig 7. Role of $x_0$ on the path.** A: A sketch of how, in hysteresis-loop bursting, $x_0$ changes the position of the z-nullcline with regards to the resting state branch. B: When the intermediate subsystem is not allowed to oscillate ($I_{rest2} = 0$), different values of $x_0$ do not change the path but only affect the velocities at which the slow variable evolves when the fast ones are in the interictal or ictal states. C: Allowing for intermediate oscillations to modulate the path ($I_{rest2} > 0$), changing $x_0$ modifies the amount of spikes in the path (more spikes when the slow variable evolves slower while in the ictal state).

## Discussion

In this work we used a minimal model for SN/SH bursting, that is the DTB bursting model, to investigate a more complex phenomenological model for seizure activity that encapsulates this type of bursting, the Epileptor model. Previous Epileptor's bifurcation analysis have focused on the two parameters $m$ and $x_0$ [51]. Our contribution here is to (i) use the map and movement on the map approach to understand the role of fast parameters on the one hand-side and slow and intermediate variables on the other, (ii) maintain all the parameters of the fast subsystem explicit in the bifurcation analysis to gain insights about their role in shaping the map, (iii) analyze previously described Epileptor's behaviors placing them in a single map together with new behaviors predicted by the generic model, highlighting the structure present in the dynamic repertoire of the Epileptor.

We started with the fast subsystem of the Epileptor and made all the parameters in it explicit. We identified those that slowly change due to coupling with slower variables, $(\mu, \bar{m})$, and analyzed their bifurcation diagram, demonstrating the topology to be equivalent to that of the DTB bursting model in the vicinity of the path for SN/SH bursting. Once obtained this map, we illustrated trajectories followed by the full Epileptor model, simulated with choices of parameters from the literature, producing SN/SH and SN/SupH hysteresis-loop bursting and depolarization block [5, 47]. In addition to already known classes, we exploited knowledge from the DTB bursting model to change the model's parameters to produce other two types of hysteresis-loop bursting: SupH/SH and SupH/SupH and to state that no other classes are possible in this map. All the classes identified so far occur in a bistability region in which the resting state fixed point is outside the limit cycle representing the ictal state. This translates in a jump in the baseline of the signal at seizure onset and offset. While this is a very common feature in human seizures, to the point that the presence of a direct current (DC) shift alone has proved indicative of the EZ [52, 52, 53], patients with focal seizures exhibits also other types of seizures, including those with no DC shift, that cannot be accounted for, at the moment, by the Epileptor [15].

With regards to the intermediate subsystem, we showed how it modulates the path, causing the fast subsystem to go through a sequence of Hopf bifurcations while moving towards seizure offset and, sometimes, altering the actual offset bifurcation. Finally we commented on the effect on the path of changing $x_0$, the Epileptor's parameter used to set the epileptogenicity of brain regions in large-scale Virtual Epileptic Patient models.

Seizures sometimes fail to terminate, evolving into Refractory Status Epilepticus (RSE), a dangerous condition difficult to treat. Two mechanisms for RSE have been proposed in the models we are dealing with and they both involve region V of the map, in which a limit cycle is the only stable attractor. This could help explain why this status is so difficult to reverse. The first mechanism (see [47] Fig 2) relies on the fact that, in a certain part of region V, the average $x_1$ activity falls below the z-nullcline so that the behavior of $z$ inverts. Now, even if the fast subsystem is not at rest the slow variable promotes movement away from the offset bifurcation. This mechanism requires an ultra-slow drift to bring the system far enough in region V to enter this regime. The second mechanism (see [15] Fig 5 D-I and Fig 14 in Appendix 1) leverages on the presence of a high level of noise in the system, which can occasionally override the slow-variable mechanism and prevent the fast subsystem to reach the offset bifurcation. It could repeatedly approach and fail to reach this bifurcation (in region III) or dwell in region V if an ultra-slow drift is present. Interestingly, only some of the 'noisy' SN/SH simulations evolved into RSE, with the rate increasing for higher levels of noise. A similar mechanism for RSE has been hypothesized in a biophysically-inspired neural mass model [54] and, given the topological equivalence between the DTB and the Epileptor

models, could be reproduced also with the latter. A deeper understanding of the transition to RSE in the model with higher noise values would require a better characterization of the corresponding stochastic differential equations [55–57]. In addition, even in the absence of noise, bursting models may display chaotic behavior at the transition from periodic bursting to spiking that may be relevant for RSE [58].

The hysteresis-loop bursting mechanism produces periodic autonomous seizures similar to those observed, for example, in *in vitro* hippocampal preparations. The statistics of the distributions of seizure onset and offset in patients, however, while justifying the use of a feedback mechanism through a slow variable for termination, give heterogeneous results for seizure initiation [59]. Seizure onset seems to be modulated by a series of mechanisms, spanning several time-scales, such as hormonal, genetic, environmental, sleep-wake cycle and behavior factors [60] that may or not act independently of the fast subsystem [61], occasionally bringing the latter in regions of the map close to seizure onset. We can thus group the wide range of timescales involved in seizure activity into three main groups, with each group pointing to different mechanisms: fast (faster than ictal length), slow (ictal length) and ultra-slow (slower than ictal length). Fast variables relate to the neuroelectric processes responsible for the generation of oscillations during the seizure (fast oscillations, spike and wave complexes, high frequency oscillations. . .). Both fast and intermediate Epileptor's variables fall into this category and have been hypothesized to reflect the activity of glutamatergic and GABAergic cells respectively [5] or mixed excitatory-inhibitory populations [13]. Slow variables, such as the Epileptor's $z$, are those responsible for seizure termination through mechanisms well represented by the feedback loop of the Epileptor model. They can be linked to a variety of processes including ionic currents, metabolic processes, alteration in the intracellular or extracellular environments, neuromodulation, but also the effects of the modulatory effect of some long-range connections, to cite a few. Finally ultra-slow variables are those, already mentioned above, responsible for bringing the system close to seizure onset, but also for transitions between seizure types or even for epileptogenesis. These variables are not modeled in the Epileptor. Both slow and ultra-slow processes are likely related to neurochemical actions [15].

Once that ultra-slow variables bring the system close to seizure onset, the transition to the ictal state can occur by crossing the bifurcation in parameter space, but also through other mechanisms. These include bifurcations as in the Epileptor, and noise-induced transitions as in the other two big families of models (biophysical or phenomenological) that have been used in the context of large-scale brain modeling for epilepsy, that is bistable models close to a subcritical H bifurcation [24, 26, 54, 62, 63] and excitable models close to a SNIC bifurcation [25, 28, 30, 38–40, 64, 65]. Other mechanisms are possible, as reviewed in [66], where we also discuss how fast-slow bursters can accommodate a mixed scenario in which the onset is brought about by noise-induced transitions and, once the system is in the ictal state the slow variable activates to bring the system across seizure offset [15, 67]. This can be achieved by setting the excitability so that the whole system is in a fixed point, as described in Section and adding noise. Depending on the interplay between the distance to the onset bifurcation and the level of noise, the system can occasionally cross the separatrix (i.e. the middle branch of fixed points) and transition into the limit cycle. It is thus possible to obtain these two onset mechanisms—bifurcations and noise-induced transitions—in the same model, allowing for their coexistence. Analyses of patients' seizures to identify signs of an impending bifurcation have brought mixed results [68, 69], with the possibility that these onset mechanisms could be patient specific [70], adding another dimension to the taxonomy of seizures based on dynamics.

In this work, we refer to the taxonomy of seizures based on dynamics, where a class is defined by its onset/offset bifurcation pair. This framework underpins the development of the Epileptor. However, other information that contributes to identifying patient-specific or seizure-specific dynamical mechanisms can also enhance the dynamic classification of seizures. These include different onset mechanisms at the neural mass level, as previously discussed, and more fine-grained spatiotemporal descriptions required for understanding the mechanisms of seizure onset, evolution, and termination [12, 13]. These taxonomies may be orthogonal to other classification systems, pathologies, or localizations [15]. However, the importance of modeling various seizures' dynamical features may vary depending on the focus of a specific investigation. For example, the intermediate subsystem of the Epileptor enhances the similarity between certain dynamotypes and observed recordings. Specifically, Perucca's class of low-frequency, high-amplitude periodic spikes (ii) [4] seems well-represented by the Epileptor's classes with a SupH onset, as shown in Fig 6. This subsystem is also crucial for understanding the rapid spread of ictal waves in the form of spike-and-wave complexes and synchronous seizure termination across brain regions [13]. However, its role is less pronounced in examining seizure propagation across different brain regions [13], leading to its exclusion from the VEP pipeline in the EPINOV clinical trial for the estimation of the epileptogenic zone (EZ) network [37]. Nevertheless, during the trial with a large cohort, we observed that the data features not included might contain valuable insights. The impact of specific modeling choices on the performance of the VEP requires further characterization [71]. This study's findings suggest a potential approach in the VEP to systematically explore the relevance of possible onset patterns, in terms of both noise-induced transitions and bifurcation, as well as specific bifurcation diagrams, in estimating EZ networks.

We here identified the two fast parameters that are slowly changed by $z$ and used part of the unfolding of the DTB singularity to guide the investigation of their bifurcation diagram. If other fast parameters of the Epileptor could be allowed to slowly change, it is possible that a similar map could be obtained for different parameters combinations, which poses a problem at the current state in trying to link the parameters of the two models. For example, in the DTB unfolding, we can obtain a topologically equivalent map also for layers with a fixed $v$ (small and positive). One possibility to start a more robust mapping between the models would be identifying a DTB singularity in the Epileptor. Interestingly, this singularity seems to play a crucial role in the organization of bifurcation diagrams of several neural and neural mass models. [72] have demonstrated that conductance based models for neurons contain such a singularity, while [73] identified the DTB point in two physiologically inspired neural mass models that have been used in the context of seizures modeling: the Jansen-Rit model and the Wendling-Chauvel. The identification of such singularity, thus, could prove to be a useful tool in analyzing a variety of models and possibly link physiological and phenomenological variables [10]. Another advantage of such an approach is that it would help identify parameters of the Epileptor model that, if allowed to slowly change, could produce other forms of bursting present in the DTB model, for example those without a DC shift that are absent from the current Epileptor's map. A lack of DC shift has been observed in some patients with focal seizures, even though this doesn't rule out completely the possibility that the underlying dynamical system is going through a bifurcation exhibiting a jump in the baseline [14], but the most known type of seizures with this characteristic is 'absence'. While the Epileptor was designed based on data from focal seizures, the possibility to extend it to generalized seizures in a unifying framework, on the lines of earlier modeling works [9, 74], is intriguing. Future work may address an extension of the Epileptor to these other bursting classes by making the fast subsystem similar to that of the DTB bursting model and carefully tuning the parameters to guarantee proper coupling with the intermediate subsystem.

## Methods

### Model

The DTB bursting model presented in Eq 2 is a simplification of the original model from [45] that only holds in this portion of the unfolding, in which the two versions of the model behave similarly with regards to bursting. The differences are three. (i) The original model used a 2D map given by the surface of a sphere of small radius centered around the DTB singularity, while here, with the only goal of simplifying the description of the model, we used a portion of the unfolding on a layer obtained with a small and fixed $\mu_2$. (ii) As a consequence of the previous choice, we could use simple segments as paths for bursting, whereas in the original model the paths were arcs of great circles. (iii) For the slow dynamics, the original model uses a more complex description that ensures that the model works even if the resting state is inside the limit cycle, which never occurs in this portion of the map.

As it is used only with an explanatory intent, Fig 1B is produced with the original model.

### Bifurcation analysis

Bifurcation curves in Fig 1A are obtained using Matcont [75]. The bifurcation analysis of the Epileptor has been done analytically and using the symbolic Matlab toolbox with regards to local bifurcations.

For the SH bifurcation we performed simulations of the fast subsystem (using Matlab function 'ode45') for the different combinations of parameters values as shown in Fig 4, using the fixed point that is not the resting state plus $\epsilon = 0.05$ as initial conditions. For the Epileptor, we simulated 600 s, removed the first 300 s to avoid the transient behavior and used the remaining to compute the amplitude and frequency of $x_1$. For the amplitude we took the difference between the maximum and minimum of the timeseries; for the frequency we used Hann window and then applied Discrete Fourier Transform. We computed amplitude and frequency of the limit cycle for the DTB model with the same procedure (as in [45]), simulating 2000 s and removing the first 500 s, with $\epsilon = 0.0005$.

**Simulations.** All simulations are performed without noise, using Matlab function 'ode15s' with maximum integration step 0.1. The Epileptor is usually used with low levels of noise that do not alter the dynamics but improve the realism of the simulations. However stronger levels of noise could potentially alter the dynamical repertoire of the model and further analysis is needed to characterize this scenario, especially given the possibility of using the Epileptor with noise-induced transition onsets as described in the Discussion.

Parameters of the model are set as in [5] unless otherwise stated, except for $x_0 = -2$ for improved visualization purposes. As described in Fig 7, this does not alter the onset/offset patterns that are the main focus of this work.

## Supporting information

**S1 Folder. Code.** The folder contains the Matlab code used to generate the figures. (ZIP)

## Author Contributions

**Conceptualization:** Maria Luisa Saggio.

**Formal analysis:** Maria Luisa Saggio.

**Funding acquisition:** Viktor Jirsa.

**Software:** Maria Luisa Saggio.

**Supervision:** Viktor Jirsa.

**Visualization:** Maria Luisa Saggio.

**Writing – original draft:** Maria Luisa Saggio.

**Writing – review & editing:** Maria Luisa Saggio, Viktor Jirsa.

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
