## [Decision Letter · Decision Letter 0]

4 Dec 2023

Dear Saggio,

Thank you very much for submitting your manuscript "Compact bifurcation analysis of the Epileptor" for consideration at PLOS Computational Biology.

As with all papers reviewed by the journal, your manuscript was reviewed by members of the editorial board and by several independent reviewers. In light of the reviews (below this email), we would like to invite the resubmission of a significantly-revised version that takes into account the reviewers' comments.

We cannot make any decision about publication until we have seen the revised manuscript and your response to the reviewers' comments. Your revised manuscript is also likely to be sent to reviewers for further evaluation.

Sincerely,

Peter Neal Taylor

Academic Editor

PLOS Computational Biology

Thomas Serre

Section Editor

PLOS Computational Biology

Reviewer's Responses to Questions

**Comments to the Authors:**

Reviewer #1: The paper considers the bifurcation analysis in a some neuron model (theoretical papers in a very narrow field), but pretends to improve surgical outcomes (the great practical problem). Unfortunately, the authors ignored all the progress made in investigation of epilepsy till now. It is hard to express all the problems in a single review, but I will try to mark at least some main points.

1. There are 30-40 types of epilepsy following ILAE. These types are actually different diseases. Each of them has its own brain structures involved, e.g. for absence epilepsy thalamocortical system is involved, though for the limbic one hippocampus and parahippocampal areas play the main role. Each of them has its own mechanism of seizure initiation, maintenance and termination. Each of them is characterized by signals with specific type, spectrum and seizure duration. Nothing of this was accounted in the paper. All the progress done was ignored. Therefore, I should conclude that the authors are compeltely out of topic and their text is not professional.

2. The epilepsy modeling is a long story. Let me provide a small introduction to the absence epilepsy modeling since this form is the simplest one and very well studied.

I. When discussing the mechanisms of thalamocortical interactions, main results were achieved using two rat strains: WAG/Rij and GAERS. Please, consider the following papers (and many others can also be found):

1. Meeren H, Pijn J, van Luijtelaar G, Coenen A, Lopes da Silva F. Cortical focus drives widespread corticothalamic networks during spontaneous absence seizures in rats. Journal of Neuroscience. 2002;22:1480–1495. doi: 10.1523/JNEUROSCI.22-04-01480.2002

2. Coenen A and van Luijtelaar G. Genetic animal models for absence epilepsy: a review of the WAG/Rij strain of rats. Behavioral Genetics. 2003; 33(6): 635–655. doi: 10.1023/A:1026179013847

3. Lüttjohann A, van Luijtelaar G. The dynamics of cortico-thalamocortical interactions at the transition from pre-ictal to ictal LFPs in absence epilepsy. Neurobiology of Disease. 2012;47:47. 60.doi: 10.1016/j.nbd. 2012.03.023.

4. Sysoeva M, Lüttjohann A, van Luijtelaar G, Sysoev I. Dynamics of directional coupling underlying spike-wave discharges. Neuroscience. 2016;314:75–89. doi: 10.1016/j.neuroscience.2015.11.044.

5. Sysoeva M, Vinogradova L, Kuznetsova G, Sysoev I, van Rijn C. Changes in corticocortical and corticohippocampal network during absence seizures in WAG/Rij rats revealed with time varying Granger causality. Epilepsy and Behavior. 2016;64:44–50. doi: 10.1016/j.yebeh.2016.08.009.

6. Sysoeva M V, Sitnikova E, Sysoev I V. Thalamo-Cortical Mechanisms of Initiation, Maintenance and Termination of Spike-wave Discharges at WAG/Rijrats. Zhurnal vysshei nervnoi deiatelnosti imeni I P Pavlova. 2016;66(1):103-112. doi: 10.7868/S0044467716010123

II. There is a classical mean-field model [Suffczynski P, Kalitzin S, Lopes Da Silva FH. Dynamics of non-convulsive epileptic phenomena modeled by a bistable neuronal network. Neuroscience.2004; 126(2): 467-84. doi: 10.1016/j.neuroscience.2004.03.014] which explains the SWDs using a system of 26 equations (not all of them were written explicitly in the paper, unfortunately, so you should reinvent some of them in order to use it), which is even more detailed than the proposed here model. It also includes GABA-ergic inhibition. If the authors model absence epilepsy, must indicate clearly what new the current study proposes in comparison to this model.

III. There different mechanisms modeling the transition to absence seizures were proposed previously

1. change in parameters, like in the early models by Alan Destexhe;

2. transition between two attractors in the bistable (multistable) system; the transition can occur due to an external stimulus from nervus trigeminus like it was proposed in [10.1016/j.neuroscience.2004.03.014] and Ref. [11], or due to some intrinsic processes modelled by noise like in [Taylor P, Wang Y, Goodfellow M, Dauwals J, Moeller F, Stephani U, Baier G. A Computational Study of Stimulus Driven Epileptic Seizure Abatement. Plos ONE. 2014; 9(12): e114316. https://doi.org/10.1371/journal.pone.0114316]

3. seizure together with interseizure interval can be considered as a long lasting event if the model allows this, see [Kolosov AV, Nuidel IV, Yakhno VG. Research of dynamic modes in the mathematical model of elementary thalamocortical cell. Izvestiya VUZ. Applied Nonlinear Dynamics. 2016; 24(5):72-83. doi: 10.18500/0869-6632-2016-24-5-72-83];

4. the epileptic state can be considered as a long transient, see [T. M. Medvedeva, M. V. Sysoeva, A. Lüttjohann, G. van Luijtelaar, I. V. Sysoev. Dynamical mesoscale model of absence seizures in genetic models. PLoS ONE, 2020 15(9), 239125. DOI: 10.1371/journal.pone.0239125] and [N. M. Egorov, V. I. Ponomarenko, I. V. Sysoev, M. V. Sysoeva. Simulation of epileptiform activity using network of neuron-like radio technical oscillators. Technical Physics, 2021 91(3): 519–528. DOI: 10.1134/S1063784221030063].

Nobody knows till now what explanation is right, and some of them could be possible together, but the authors should show that they are familiar with different approaches, what is their own approach and why they chose it.

Reviewer #2: Compact Bifurcation Analysis of the Epileptor by Maria Luisa Saggio and Viktor K. Jirsa

The MS deals with a key problem of modelling of mesoscopic macroscopic dynamics in the human EEG / MEG in patients with electrographic seizures: it studies and connects transitions between different types of dynamics and thus attempts to explain deterministic components of large scale brain activity. This is crucial for surgical decision making.

A key assumption underlying the work is that seizures result from local dynamical instabilities, typically from a fixed point to some kind of oscillatory behavior. A lot of modelling work rests on purely phenomenological similarity of simulated output with clinical recordings and parameters are empirically fixed to show the desired dynamics but ignoring the potential impact of other parameters. At this point, the MS offers a step forward in reducing the 5-variable epileptor model to the minimal number of variables, which is 3 for bursting, and to thoroughly analyse that minimal model for generic routes in and out of oscillatory behavior (or between two steady states) to understand the more complex 5-variable model with its larger number of parameters.

Overall the MS is of excellent technical quality and clear reasoning.

Specifically, figure 4 gives confidence that the DTB results can be used to predict epileptor dynamics and in particular different routes of transitions and how they relate to each other e.g. given "paths on the map", as the authors show.

Given the newly gained understanding of the dependence of the bifurcation structure on the slowly varying parameters, it will be easier for dynamically trained researchers to adjust the model to observations in time series.

Some comments and critique:

Different phenomenologies are illustrated based on the bifurcations but their relevance to the clinical observations is not discussed. It might good to compare the simulated seizure onset dynamics to the classes proposed by Perucca, Dubeau and Gotman (2013). Particularly, the new classes in Fig. 6.

One shortcoming of the MS is that while it refers to "seizure", "ictal", and "interictal" states with the clinical application in mind, these are interpretations only. It should state what type of asymptotic dynamics is found (for the analysis, they cut transients away). Are these period, quasiperiodic, chaotic?

A point that might be slightly misleading in the mathematical approach is the implicit assumption that the illustrated dynamics happen in one "location" and thus the choice of parameters determines the "node" dynamics of a lump of e.g. cortex and on top of this connectivity needs to be added to account for spreading. Yet, the dynamics could in principle also be the result of the coupling between potentially distant (in the extreme case trivially behaving) locations. While bifurcations of spatio-temporal models are outside the scope of the MS , it is important that this at least is mentioned.

All studies are done in the deterministic case in the absence of noise. While this allows convenient analytic treatments, the corresponding stochastic differential equations might in principle have a more complex dynamics than is expected from the noise-free case. I suggest to point this out and maybe include a citation.

I want to mention that seizure-like dynamics could in principle also originate in behaviors other than passing a slow variable through a bifurcation. E. g. it could be a noise-induced transient exploiting a complicated saddle structure of state space. I leave it to the authors but think it is fair to mention that this specific approach has not yet been conclusively shown to be the correct one.

Finally, given the good correspondence between the two models, would the authors continue to advise clinical people to work with the complex epileptor rather then with the simpler minimal model? Specifically, the simpler model model might be easier to modify if further variables are added (as per the suggestions in the Discussion).

Some remarks:

I think the piecewise model eq (5) is a brilliant tool to work with.

I didn't check the analytical results in 4.1.1

Figure 2 is a nice didactic illustration of the epileptor's working

Some suggestions:

line 82: should "deg. DTB" just be "DTB"?

line 382: "iniziation"

line 400: "epileptogenisis"

**Have the authors made all data and (if applicable) computational code underlying the findings in their manuscript fully available?**

Reviewer #1: Yes

Reviewer #2: **No: **I didn't get to see code in the downloaded submission.

PLOS authors have the option to publish the peer review history of their article (what does this mean?). If published, this will include your full peer review and any attached files.

Reviewer #1: No

Reviewer #2: **Yes: **Gerold Baier
---

## [Decision Letter · Decision Letter 1]

8 Feb 2024

Dear Saggio,

We are pleased to inform you that your manuscript 'Bifurcations and bursting in the Epileptor' has been provisionally accepted for publication in PLOS Computational Biology.

Although Reviewer 1 still had serious concerns with the study, I am of the opinion that your response was adequate. As a specific example, where reviewer 1 states "The universal epileptor they propose dose not meet any scientific criteria, because there is no epilepsy in general...", I considered your response "...the Epileptor model was built to simulate focal seizures data from patients undergoing pre-surgical evaluation..." appropriate. Other examples also exist in the response. Therefore I am of the opinion your work should be provisionally accepted for publication. Further comments from both reviewers can be found later in this email.

Best regards,

Peter Neal Taylor

Academic Editor

PLOS Computational Biology

Thomas Serre

Section Editor

PLOS Computational Biology

Reviewer's Responses to Questions

**Comments to the Authors:**

Reviewer #1: Unfortunately the authors could not fix the main problem of the study. The universal epileptor they propose dose not meet any scientific criteria, because there is no epilepsy in general and the mechanisms of epilepsy are very different for different forms. Modeling epilepsy in general is like modeling high temperature for all patients from cancer IV stage to influenza ones.

The answers of the authors clearly show that the model they propose in the current framework cannot be used for any real purpose (practical value) and cannot meet any real biological measurements (theoretical value). I do not see any reason to publish valueless paper.

Reviewer #2: The authors have addressed my issues.

**Have the authors made all data and (if applicable) computational code underlying the findings in their manuscript fully available?**

Reviewer #1: Yes

Reviewer #2: Yes

PLOS authors have the option to publish the peer review history of their article (what does this mean?). If published, this will include your full peer review and any attached files.

Reviewer #1: No

Reviewer #2: No

---

## [Editor Report · Acceptance letter]

29 Feb 2024

PCOMPBIOL-D-23-01749R1 

Bifurcations and bursting in the Epileptor

Dear Dr Saggio,

I am pleased to inform you that your manuscript has been formally accepted for publication in PLOS Computational Biology. Your manuscript is now with our production department and you will be notified of the publication date in due course.

With kind regards,

Anita Estes
